# Toll-like Receptor 2 Mediated Immune Regulation in Simian Immunodeficiency Virus-Infected Rhesus Macaques

**DOI:** 10.3390/vaccines11121861

**Published:** 2023-12-17

**Authors:** Nongthombam Boby, Kelsey M. Williams, Arpita Das, Bapi Pahar

**Affiliations:** 1Division of Comparative Pathology, Tulane National Primate Research Center, Covington, LA 70433, USA; nboby@txbiomed.org (N.B.); kwilli30@tulane.edu (K.M.W.); 2Division of Microbiology, Tulane National Primate Research Center, Covington, LA 70433, USA; arpita.das@nih.gov; 3School of Medicine, Tulane University, New Orleans, LA 70118, USA

**Keywords:** chemokines, cytokines, gut, innate immunity, peripheral blood, rhesus macaque, SIV, toll-like receptor 2

## Abstract

Toll-like receptors (TLRs) are crucial to the innate immune response. They regulate inflammatory reactions by initiating the production of pro-inflammatory cytokines and chemokines. TLRs also play a role in shaping the adaptive immune responses. While this protective response is important for eliminating infectious pathogens, persistent activation of TLRs may result in chronic immune activation, leading to detrimental effects. The role of TLR2 in regulating HIV-1 infection in vivo has yet to be well described. In this study, we used an SIV-infected rhesus macaque model to simulate HIV infection in humans. We evaluated the plasma of the macaques longitudinally and found a significant increase in the soluble TLR2 (sTLR2) level after SIV infection. We also observed an increase in membrane-bound TLR2 (mb-TLR2) in cytotoxic T cells, B cells, and NK cells in PBMC and NK cells in the gut after infection. Our results suggest that sTLR2 regulates the production of various cytokines and chemokines, including IL-18, IL-1RA, IL-15, IL-13, IL-9, TPO, FLT3L, and IL-17F, as well as chemokines, including IP-10, MCP-1, MCP-2, ENA-78, GRO-α, I-TAC, Fractalkine, SDF-1α, and MIP-3α. Interestingly, these cytokines and chemokines were also upregulated after the infection. The positive correlation between SIV copy number and sTLR2 in the plasma indicated the involvement of TLR2 in the regulation of viral replication. These cytokines and chemokines could directly or indirectly regulate viral replication through the TLR2 signaling pathways. When we stimulated PBMC with the TLR2 agonist in vitro, we observed a direct induction of various cytokines and chemokines. Some of these cytokines and chemokines, such as IL-1RA, IL-9, IL-15, GRO-α, and ENA-78, were positively correlated with sTLR2 in vivo, highlighting the direct involvement of TLR2 in the regulation of the production of these factors. Our findings suggest that TLR2 expression may be a target for developing new therapeutic strategies to combat HIV infection.

## 1. Introduction

The hallmark of HIV infection is the loss of CD4+ T cells, immune suppression, disruption of gut epithelial homeostasis, generation of viral reservoirs, and generalized acquired immunodeficiency syndrome [1,2]. In the early stages of HIV infection, there is a rapid replication of the virus in the gut-associated lymphoid tissue. This process leads to a considerable reduction in CD4+ T cells and the loss of intestinal barrier function. As a result, intestinal bacteria can enter the bloodstream, triggering a state of chronic systemic immune activation that perpetuates HIV replication and increases the likelihood of progression to AIDS [3,4,5,6,7,8,9,10]. The innate immune response is the first line of defense against pathogens. Toll-like receptors (TLRs) are a group of pattern recognition receptors that identify pathogen-associated molecular patterns (PAMPs), including PAMPs from bacteria [11,12,13,14], viruses [15,16,17,18], fungi [19], and parasites [20]. They are critical in the innate immune response [21,22]. The activation of NF-κB, IRFs, or MAP kinase occurs by utilizing adapter proteins like MyD88 and TRIF (Toll/interleukin-1 receptor domain-containing adaptor protein inducing interferon β) by TLRs. This effectively regulates cytokines, chemokines, and interferons (IFNs), which provides exceptional protection against microbial infections [23]. Microbial or microbial product translocation due to dysregulation of the gut epithelial barrier in HIV infection could constantly stimulate various TLRs, resulting in persistent TLR signaling and immune activation. The continual engagement of TLRs could lead to secondary immunosuppression to neutralize chronic proinflammatory responses and eventually favor HIV immunopathogenesis. The differential expression of TLRs in different pathological conditions of HIV/simian immunodeficiency virus (SIV) infection has been reported [24,25]. The trans-membrane domain of the HIV envelope directly interacts with TLR2 and attenuates TNF-α, IL-6, and MCP-1 secretions of macrophages in in vivo and in vitro models [26]. Neutralizing the TLR2 pathway in primary female genital epithelial cells prevents the induction of proinflammatory cytokines and epithelial barrier breakdown by gp120, depending on the presence of heparan sulfate [27]. HIV-infected patients displayed enhanced TLR2 expression, increasing viral replication and TNF-α response [28]. TLR2 agonists suppress HIV-1 replication in monocyte-derived macrophages by regulating the production of IL-10 and CCL3, CCL4, and CCL5 expression [29]. The TLR2 ligand increases virus transfer from immature monocyte-derived dendritic cells to autologous CD4+ T cells in vitro [30]. In contrast, HIV-1 infection in dendritic cells (DCs) and DC-mediated virus transmission to CD4+ T cells is reduced upon TLR4 stimulation [30]. Soluble TLR2s (sTLR2) are considered negative regulators of TLR2-mediated immune responses, where sTLR2 can compete with TLR2 agonists [11]. The precise involvement of TLR2 in regulating HIV-1 infection in vivo remains an enigma. Thus, a well-designed, controlled natural history study employing a nonhuman primate model (NHP) is indispensable to elucidate the dynamics of TLR2 during SIV infection. The NHP-SIV model has been widely used to study the pathogenesis and treatment of HIV due to their similarities in immune responses and pathogenesis with human HIV disease. We hypothesized that both sTLR2 and membrane-bound TLR2 (mb-TLR2) may trigger chronic immune activation by regulating the production of inflammatory cytokines and chemokines and viral replication and HIV-1 disease progress. Utilizing a non-human primate model, our study extensively examines TLR2 expression in tissues and how it relates to cytokine/chemokine shifts during SIV infection.

## 2. Materials and Methods

### 2.1. Ethics Statement, SIV Infection, and Sample Collection

Thirteen Indian Rhesus macaques (RhMs, *Macaca mulatta*) of both sexes (six females and seven males) between 6.4 and 8.3 years of age were housed at Tulane National Primate Research Center (TNPRC) in accordance with the NIH guidelines. The subjects were cared for, and their well-being was ensured by following the standard animal procedures humanely [31]. This study was approved by the Tulane University Institutional of Laboratory Animal Care and Use Committee (IACUC). At the start of this study, animals were negative for SIV, simian T cell leukemia virus type 1, and type D retrovirus antibodies. Three RhMs were used for in vitro TLR2 agonist study and remained as SIV-uninfected controls. The remaining ten RhMs were infected with 100 TCID_50_ pathogenic SIV_MAC_251 intravenously, representing a common route of human HIV transmission.

The macaques were housed in indoor facilities with biosafety level 2 measures and environmental controls. Each macaque was housed individually and provided with a diet designed for primates, feeding enrichment, and unlimited access to water. All subjects were monitored twice daily for any signs of pain, distress, or disease. Blood collections, tissue collections, physical exams, or virus inoculation were performed under anesthesia using ketamine hydrochloride (10 mg/kg of body weight (BW)) or tiletamine hydrochloride/zolazepam (Telazol, Zoetis, Parsippany, NJ, USA) (5–8 mg/kg of BW) given intramuscularly [32,33]. Several samples were collected at various time points, including 0, 7, 14, 21, 40, 60, 70, 90, 120, 150, and 180 days post-infection (dpi) to perform virological and immunological assays (Figure 1A).

### 2.2. Plasma Isolation

Plasma was isolated and stored at −80 °C after centrifugation of EDTA-anticoagulated blood at 2500 rpm for 10 min. Frozen plasma samples were thawed to perform various assays, including viral RNA quantification, ELISAs, and quantification of different cytokines and chemokines.

### 2.3. Hematology

Complete blood counts (CBCs) for hematology were analyzed using a Sysmex XT2000i analyzer (Sysmex Corporation, Kobe, Japan) on freshly drawn EDTA-anticoagulated blood.

### 2.4. Quantitative SIV Plasma Virus Load (PVL)

Viral RNA was isolated from plasma samples using the Maxwell RSC Viral Total Nucleic Acid Purification Kit on the Maxwell 48 RSC instrument (Promega, Madison, WI, USA). RNA was reverse transcribed and amplified using the TaqMan Fast Virus 1-Step Master Mix qRT-PCR kit (Thermo Fisher Scientific, Waltham, MA, USA) on the LightCycler (LC) 480 or LC96 instrument (Roche, Indianapolis, IN, USA). Quantified RNA was transcribed from the p239gag_Lifson plasmid (kindly provided by Dr. Jeffrey Lifson, Frederick National Laboratory). The final PCR reaction mixtures contained 150 ng random primers (Promega, Madison, WI, USA), 600 nM of each primer, and 100 nM probe. Primers and probe sequences are as follows: forward primer: 5′-GTC TGC GTC ATC TGG TGC ATT C-3′, reverse primer: 5′-CAC TAG GTG TCT CTG CAC TAT CTG TTT TG-3′, and probe: 5′-6-carboxyfluorescein-CTT CCT CAG TGT GTT TCA CTT TCT CTT CTG CG-BHQ1-3′. The reactions were cycled with the following conditions: 50 °C for 5 min, 95 °C for 20 s, followed by 50 cycles of 95 °C for 15 s and 62 °C for 1 min. Quantification cycle data and a serial dilution of a highly characterized custom RNA transcript of the SIVgag sequence were used to interpolate mean SIVgag RNA copies per reaction. The limit of this assay was 60 SIV RNA copies/mL of plasma [34].

### 2.5. Isolation of Peripheral Blood Mononuclear Cells (PBMCs)

PBMCs were isolated from heparinized whole blood by Ficoll density gradient centrifugation [35,36]. Briefly, the plasma from the whole blood was separated by centrifugation at 2500 rpm for 10 min. The pellet was resuspended with PBS and slowly layered over a layer of Ficoll in a 50 mL conical tube. After centrifugation at 1800 rpm for 40 min with no brake, the buffy coat containing PBMC was separated gently without disturbing the other layers. The PBMC was washed twice with 2% fetal bovine serum (FBS)-PBS and resuspended in RPMI-1640 with 10% FCS (Cambrex, Walkersville, MD, USA), supplemented with 10 mM HEPES, 2 mM L-glutamine, 100 U/mL penicillin, and 100 µg/mL streptomycin (BioWhittaker, Lonza, Walkersville, MD, USA) (referred to as complete media onward).

### 2.6. Isolation of Jejunal Lamina Propria Leukocytes (LPL)

The jejunal LPL was isolated from the jejunum by collagenase treatment followed by Percoll density gradient centrifugation [8,33,37]. The jejunal pieces were resuspended with HBSS containing 5 mM EDTA and 1 mM DDT, incubated in a shaker platform, and shaken at 300 rpm, 37 °C for 30 min. The tissue suspension was strained through a screen cup strainer to filter the epithelial cells out. The tissue pieces on the strainer were further minced and digested by shaking them at 300 rpm, 37 °C, for 1 h with type II collagenase (60 U/mL) (Sigma-Aldrich, St. Louis, MO, USA) prepared in the complete media. The tissue enzyme suspension was diluted with a similar volume of cold complete media to stop the enzyme activity and passed through a 16-gauge feeding needle. The larger clumps and undigested tissue were filtered using a nylon biopsy bag (Fisher Scientific, Hampton, NH, USA). The filtrate was spun at 1700 rpm for 10 min at 4 °C. The cell pellet was reconstituted in complete media. The cell suspension was layered over the 35% and 60% percoll gradients and centrifuged at 1800 rpm with no brake for 30 min at 4 °C. The interface layer between the two percoll layers was collected, washed, resuspended with complete media, and counted for further experiments.

### 2.7. Isolation of Cells from Lymph Nodes (LNs)

Peripheral LNs were processed, as discussed previously [35,36]. In brief, the tissue was collected in complete media, taken in the lab, cut into small pieces, and transferred onto a 70 µm strainer placed in a Petri dish. The pieces were mashed through the strainer using a syringe plunger to make a single-cell suspension. The cell suspension was washed after adding 20 mL of complete media and centrifuged at 1300 rpm for 7 min. The washed cell pellet was resuspended with complete media and counted for further experiment. The cell viabilities of isolated PBMCs, jejunum lamina propria leukocytes, and lymph nodes leukocytes were counted with the Nexcelom auto cell counter (Nexcelom Bioscience LLC, Waltham, MA, USA) using acridine orange/propidium iodide dye.

### 2.8. In Vitro TLR2 Stimulation

PBMCs from three healthy, SIV-uninfected RhMs were isolated. Freshly isolated PBMCs at a concentration of 1–1.5 × 10^6^/mL cells were cultured in complete media in the presence of soluble protein A from *Staphylococcus aureus* (final concentration 100 μg/mL) (Sigma-Aldrich) as TLR2 agonist [38,39] in a 24-well tissue culture plate. PBMCs stimulated with 1× PBS were used as controls for each animal. The cells were maintained at 37 °C for 48 h in a 5% CO_2_ incubator. The supernatant was collected after the incubation and stored at −80 °C.

### 2.9. Quantification of Cytokines and Chemokines in Plasma and Cell Culture Supernatant

The quantification of sixty-one cytokines/chemokines was performed using the U-plex biomarker NHP 61plex (Meso Scale Diagnostics; MSD, Rockville, MD, USA). The manufacturer’s instructions were followed with minor modifications while utilizing plasma and cell culture supernatant samples [32,33]. The list of 39 cytokines includes FLT3L (FMS-like tyrosine kinase 3 ligand), G-CSF (granulocyte colony-stimulating factor), GM-CSF (granulocyte-macrophage colony-stimulating factor), I-309, IFN-α2a, IFN-γ, IL-1α (interleukin-1α), IL-1β, IL-1RA (interleukin-1 receptor antagonist), IL-2, IL-2Rα, IL-4, IL-5, IL-6, IL-7, IL-8, IL-9, IL-10, IL-12, IL-12p70, IL-13, IL-15, IL-16, IL-17A, IL-17A/F, IL-17B, IL-17C, IL-17D, IL-17F, IL-18, IL-22, IL-23, M-CSF (macrophage colony-stimulating factor), MIF (macrophage migration inhibition factor), TNF-α (tumor necrosis factor-α), TNF-β, TPO (thrombopoietin), TRAIL (TNF-related apoptosis-inducing ligand), and YKL-40 (chitinase-3-like protein 1). The list of 22 chemokines tested in this assay is CTACK (C-c motif chemokine ligand 27), eotaxin-1, eotaxin-2, eotaxin-3, ENA-78, Fractalkine, GRO-α, IP-10 (IFN-γ-inducible protein 10), I-TAC (Interferon-inducible T cell alpha chemoattractant), MCP-1 (monocyte chemotactic protein-1), MCP-2, MCP-3, MCP-4, MDC (macrophage-derived chemokine), MIP-1α (macrophage inflammatory protein 1α), MIP-1β, MIP-3α, MIP-3β, MIP-5, SDF-1α (stromal cell-derived factor-1 alpha), TARC (thymus and activation-regulated chemokine), and VEGF-α (vascular endothelial growth factor-α). In brief, the plates were coated with biotinylated capture antibodies, followed by incubation with calibrator standards and diluted plasma samples. After washing, the detection antibody was added, and the plate was incubated. Finally, the sample was read on an MSD microplate reader to establish cytokine and chemokine concentrations.

### 2.10. Quantification of Microbial Translocation by Measuring Intestinal Fatty Acid Binding Protein (I-FABP)

Plasma I-FABP levels from all RhMs were measured using a FABP kit (DuoSet ELISA, R&D Systems Inc., Minneapolis, MN, USA) in duplicates following the manufacturer’s instructions, with the detection limit ranging from 31.2 to 2000 pg/mL. For quantification of plasma I-FABP level, a standard curve with known FABP concentrations was generated. Nonlinear regression using a sigmoidal dose–response variable slope model was used to interpolate concentrations from the standard curve.

### 2.11. Soluble CD14 (sCD14) Marker Quantification

Plasma CD14 levels were measured at different pre- and post-SIV infection time points using a quantitative sCD14 sandwich ELISA (Human CD14, DuoSet ELISA, R&D Systems, Inc., Minneapolis, MN, USA) in duplicates following the manufacturer’s instruction. This assay’s detection limit was 62.5–4000 pg/mL.

### 2.12. Quantification of Plasma REG3A

REG3A (Regenerating islet-derived protein 3 alpha) concentration in plasma was measured at different pre- and post-SIV infection time points using a quantitative sandwich Reg3A ELISA (Human Reg3A, DuoSet ELISA, R&D Systems, Inc., Minneapolis, MN, USA). Samples were run in duplicates, and the data were interpolated from the standard curve using a sigmoidal dose–response variable slope model. The sensitivity of the assay ranged from 15.6 to 1000 pg/mL.

### 2.13. Quantification of Soluble TLR2 (sTLR2) in Plasma

Monkey TLR2 ELISA kit was used to quantify plasma sTLR2 concentration according to the manufacturer’s protocol (Lifespan Bioscience Inc., Shirley, MA, USA). Briefly, the frozen plasma was thawed and centrifuged at 2000 rpm for 10 min. The standards and samples prepared in two-fold dilution were added into the designated wells and incubated overnight at 4 °C. The sample solution was aspirated and washed the next day. Biotinylated detection antibody and HRP-conjugated solution were sequentially added to each well and incubated at 37 °C for 60 min and 30 min, respectively. Finally, TMB substrate solution was added, and the plate was read at the 450 nm optical density after stopping the reaction. Nonlinear regression using a sigmoidal dose–response variable slope model was used to interpolate plasma sTLR2 concentration from the known standard curve.

### 2.14. Flow Cytometry Analysis of Membrane-Bound TLR2 (mb-TLR2) Expression

Freshly isolated PBMCs, jejunum LPL, and LN cells were stained for flow cytometry following the protocol reported earlier [8,35,37]. In brief, 1–1.5 × 10^6^ mL cells were stimulated with Staphylococcal enterotoxin (SEB, Toxin Technologies, Sarasota, FL, USA) at 5 µg/mL and kept overnight in a 5% CO_2_ incubator at 37 °C. The stimulated cells were adjusted in 100 µL flow wash buffer (0.5% BSA-PBS) and stained with an appropriate amount of aqua fluorescent reactive dye live/dead stain (Life Technologies, Carlsbad, CA, USA) for 10 min at 37 °C and subsequently with fluorochrome-conjugated antibodies (Appendix A) for 25 min at room temperature. Stained cells were washed with wash buffer and fixed with 1X BD stabilizing fixative buffer (BD Biosciences, Franklin Lakes, NJ, USA). Cells were kept in the dark at 4 °C, and flow acquisition was performed within 24 h of staining. The flow cytometric acquisition was performed using the Becton, Dickinson Fortessa instrument (BD Biosciences, Franklin Lakes, NJ, USA. The acquired cells were analyzed using FlowJo software version 10.9 (FlowJo, Ashland, OR, USA). Only the singlets and live cell populations were considered to analyze all the markers. Each sample was subjected to at least 50,000 events for acquisition. To analyze the absolute count of CD4 and NK cells in peripheral blood, we used data from CBCs and flow cytometry for CD4+ (CD45+CD3+CD4+) T cells and NK (CD45+CD3–CD8+) cell percentages.

### 2.15. Statistical Analyses

GraphPad Prism was used for all statistical analyses and graph generation (v10.1.1., GraphPad Software, Boston, MA, USA). One-way repeated ANOVA analysis was performed to identify if there were any significant differences between multiple time points. Tukey’s multiple comparison tests compared different time points for statistically significant changes. A paired T-test was used to identify any significant differences in the cytokine or chemokine production in the in vitro culture experiment. A two-tailed Pearson correlation analysis was used to measure the correlation between parameters. The definition of *p* values is as follows: nonsignificant, *p* > 0.05; *, *p* ≤ 0.05; **, *p* ≤ 0.01; ***, *p* ≤ 0.001; ****, *p* ≤ 0.0001.

## 3. Results

### 3.1. Dynamics of Plasma Viral Load and Absolute Peripheral CD4 Count

All the ten SIV-infected RhMs had detectable PVL, where peak plasma viral replication was detected between 14 and 21 days post-infection (dpi) with a range of log_10_ 5.9–8.2 (Figure 1B). Subsequently, viral loads declined but increased again during the later phase of SIV infection at 70 dpi (log_10_ 6.0–7.6) and remained high at 180 dpi, which was the end phase of this study (log_10_ 5.0–8.1). The absolute CD4 count in the peripheral blood significantly decreased from day 14 post-SIV infection (mean ± SE, 605 ± 96 cells/µL of blood, *p* ≤ 0.05) and remained low throughout the study period when compared to the preinfection levels (1247 ± 108 cells/µL of blood) (Figure 1C). We observed no significant differences in PVL and absolute CD4 count between male and female subjects in our study.

### 3.2. An Increased Level of sTLR2 Was Detected in the Plasma of SIV-Infected RM

The concentration of plasma sTLR2 was assessed at various time points after SIV infection. During the early acute stage of infection at 14 dpi, the plasma sTLR2 concentration was significantly increased (mean ± SE: 2167 ± 324.6 pg/mL, *p* = 0.036) compared to the pre-infection time point (1412 ± 304.7) (Figure 2A). The sTLR2 concentration remained higher at 21, 40, 60, and 90 dpi, but these levels were not significantly different when compared to the 0-day time point. However, during the late chronic stage of infection at 120 and 180 dpi, the sTLR2 concentration was elevated considerably (2330 ± 302.9 pg/mL, *p* = 0.036 and 4401 ± 684.3 pg/mL, *p* = 0.018, respectively) when compared to the 0-day time point (Figure 2A).

### 3.3. Dynamics of Cytokine Expression during SIV Infection

The plasma cytokine concentrations were quantified from all ten RhMs for all the time points. Out of a total of thirty-nine cytokines, we noted that eight different cytokines, namely IL-15, TPO, IL-18, IL-13, FLT3L, IL-1RA, IL-9, and IL-17F, significantly altered at one or most time points when compared to their baseline data and have a correlation in their expression with plasma sTLR2 levels (Figure 2B–I). IL-15, a positive regulator of several immune cellular processes, such as cell maturation and T and NK cell activation, was significantly increased at 180 dpi (20.7 ± 2.8 pg/mL, *p* = 0.046) compared to baseline (9.3 ± 0.7) (Figure 2B). TPO, a lineage-specific cytokine responsible for megakaryocyte proliferation and maturation and the promotion of circulatory platelets, [40] was also significantly upregulated at 180 dpi (266.8 ± 20.4 pg/mL, *p* = 0.023) compared to baseline (191.4 ± 18.4) (Figure 2C).

IL-18, a proinflammatory cytokine that modulates both innate and adaptive immunity [32,41], significantly increased at 14 (378.4 ± 74.9 pg/mL, *p* = 0.028), 21 (212.9 ± 28.6 pg/mL, *p* = 0.013), and 90 dpi (134.3 ± 13.4 pg/mL, *p* = 0.005) compared to the baseline time point (87.9 ± 8.1 pg/mL) (Figure 2D). IL-13 promotes B cell proliferation, differentiation, and immunoglobulin class switching [42]. It also upregulates low-affinity IgE receptors in B cells and monocytes. At 180 dpi (14.4 ± 1.8 pg/mL, *p* = 0.041), there was a significant increase in the IL-13 expression compared to baseline (4.2 ± 2.4) (Figure 2E). FLT3L is essential in the growth and development of NK and dendritic cells [43]. The plasma FLT3L concentration increased significantly at 14 (53.0 ± 8.2 pg/mL, *p* = 0.006), 120 (34.7 ± 4.4 pg/mL, *p* = 0.009), and 180 dpi (82.7 ± 13.5 pg/mL, *p* = 0.011) compared to the preinfection time point (16.9 ± 1.8 pg/mL) (Figure 2F). IL-1RA is a naturally occurring cytokine that functions as a competitive inhibitor of proinflammatory cytokines: IL-1α, Il-1β. It also regulates monocyte activation by binding with IL-1R1 [44,45], which is upregulated at 60 (512.8 ± 68.4 pg/mL, *p* = 0.022), 120 (697.6 ± 120.3 pg/mL, *p* = 0.037), and 180 dpi (2054 ± 437.4 pg/mL, *p* = 0.026) compared to the preinfection time point (211.5 ± 19.3) (Figure 2G). IL-9 is a cytokine with pleiotropic effects produced by various cells, including Th9 cells. It plays a crucial role in regulating inflammation, proliferation of Th17 cells, production of IgG1 and IgE by B cells, proliferation of goblet cells, and immunity against parasites and cancer [33]. At 90 dpi, the level of IL-9 significantly decreased to 0.01 ± 0.008 pg/mL (*p* = 0.029) from the baseline level of 0.05 ± 0.001 pg/mL (Figure 2H). IL-17F, a cytokine of the IL-17 family, is essential in regulating inflammatory responses and maintaining mucosal barrier function [46]. The plasma IL-17F level upregulated after SIV infection, but only a significant increase was detected at 21 dpi (89.7 ± 20.6 pg/mL, *p* = 0.034) when compared to baseline (33.7 ± 11.2) (Figure 2I).

### 3.4. Dynamics of Chemokine Expression during SIV Infection

After SIV infection, a significant difference was detected in nine different chemokines (IP-10, MCP-1, MCP-2, ENA-78, GRO-α, I-TAC, Fractalkine, SDF-1α, and MIP-3α), which also correlates with the sTLR2 expression (Figure 3). IP-10 is secreted by multiple lymphoid and nonlymphoid cells to regulate chemotaxis, cell apoptosis, cell growth inhibition, and angiogenesis [47]. The plasma IP-10 concentration was significantly upregulated at 14 (1734 ± 312 pg/mL, *p* = 0.030), 21 (2485 ± 292 pg/mL, *p* = 0.0004), 60 (1517 ± 160 pg/mL, *p* = 0.004), 90 (1718 ± 267 pg/mL, *p* = 0.020), 120 (1692 ± 275 pg/mL, *p* = 0.042), and 180 dpi (4340 ± 802 pg/mL, *p* = 0.014) compared to the baseline time point (570 ± 61 pg/mL, Figure 3A). Similar to IP-10, MCP-1 and MCP-2 expression was also upregulated after infection. The MCP-1 level was significantly upregulated at 14 (137.1 ± 10.8 pg/mL, *p* = 0.002), 21 (119.2 ± 6.2 pg/mL, *p* = 0.0003), 90 (128.8 ± 9.0 pg/mL, *p* = 0.02), and 120 dpi (132.3 ± 12.0 pg/mL, *p* = 0.008) compared to the preinfection time point (81.4 ± 4.9 pg/mL, Figure 3B). The MCP-2 level also significantly increased at 14 (6.5 ± 0.9 pg/mL, *p* = 0.009), 21 (7.2 ± 1.1 pg/mL, *p* = 0.018), 60 (10.8 ± 2.2 pg/mL, *p* = 0.023), and 120 dpi (9.4 ± 1.9 pg/mL, *p* = 0.028) compared to baseline (2.7 ± 0. pg/mL, Figure 3C). ENA-78, a member of the CXC chemokines, which acts as a potent neutrophil chemoattractant and activator [48], changed significantly after SIV infection (Figure 3D). However, there was no significant difference detected in ENA-78 level between any post-infection time point when compared to the preinfection time point. Similar to MCP-1 and MCP-2, GRO-α, which is responsible for neutrophil recruitment during inflammation [49], significantly increased at 14 dpi (258.1 ± 49.7 pg/mL, *p* = 0.043) compared to baseline (102.0 ± 20.2 pg/mL, Figure 3E).

ITAC, a potent chemoattractant of IL-2-activated T cells was also significantly upregulated at 14 (740.6 ± 113.9 pg/mL, *p* = 0.005), 21 (902.8 ± 131.9 pg/mL, *p* = 0.001), 60 (455 ± 55.0 pg/mL, *p* = 0.003), 90 (536.3 ± 35.3 pg/mL, *p* < 0.0001), 120 (746.2 ± 90.1 pg/mL, *p* = 0.003), and 180 dpi (1441 ± 233.4 pg/mL, *p* = 0.004) compared to the preinfection time point (148.6 ± 19.8 pg/mL) (Figure 3F). Fractalkine, a chemokine with pleotropic function like cell adhesion, cell migration, chemotaxis, and cell survival [50], was significantly upregulated very early at 14 dpi (15.0 ± 1.4 ng/mL, *p* = 0.006) compared to baseline (9.2 ± 0.6 ng/mL). However, no significant change in the Fractalkine level was detected during the chronic phase of infection (Figure 3G). Interestingly, plasma SDF-1α, a vital chemokine responsible for cell homing, cell migration, and cell regeneration, remained stable except at 120 dpi, where the level remained significantly lower (2085 ± 624.5 pg/mL, *p* = 0.0005) when compared to the preinfection time point (2734 ± 644.9 pg/mL) (Figure 3H). The antimicrobial and anti-HIV chemokine MIP-3α, which is also responsible for inducing innate immune defense, was significantly increased at 180 dpi (13.6 ± 2.5 pg/mL, *p* = 0.041) compared to the baseline values (4.7 ± 0.3 pg/mL, Figure 3I).

### 3.5. Correlation of Plasma Viral Load with Plasma sTLR2, IL-18, and FLT3L Concentration

The correlation between plasma sTLR2 and viremia was analyzed to determine whether the increased sTLR2 level was associated with the increasing PVL. A positive correlation was observed between plasma sTLR2 and SIV plasma viral load (r = 0.31, *p* = 0.008, Figure 4A). A statistically significant positive correlation between PVL and plasma IL-18 (r = 0.37, *p* = 0.005, Figure 4B) or FLT3L (r = 0.45, *p* = 0.0006, Figure 4C) concentrations were also observed.

### 3.6. Evidence of a Positive Correlation between Plasma sTLR2 and Different Cytokines/Chemokines

The correlation of plasma sTLR2 with individual cytokine concentration was evaluated over time in all 10 SIV-infected RhMs (Figure 4D). A positive correlation was observed between sTLR2 and IL-18 (r = 0.29, *p* = 0.015), sTLR2 and IL-1RA (r = 0.81, *p* < 0.0001), sTLR2 and IL-15 (r = 0.69, *p* < 0.0001), sTLR2 and IL-13 (r = 0.29, *p* = 0.014), sTLR2 and IL-9 (r = 0.42, *p* = 0.0003), sTLR2 and TPO (r = 0.43, *p* = 0.0002), sTLR2 and FLT3L (r = 0.62, *p* < 0.0001), and sTLR2 and IL-17F (r = 0.27, *p* = 0.023). There was no significant correlation between sTLR2 and other cytokines tested in SIV-infected RhMs.

Similarly, sTLR2 and individual chemokine concentration were evaluated over time in all 10 SIV-infected RhMs (Figure 4E). A positive correlation was detected between sTLR2 and IP-10 (r = 0.74, *p* < 0.0001), sTLR2 and MCP-1 (r = 0.84, *p* < 0.0001), sTLR2 and MCP-2 (r = 0.50, *p* < 0.0001), sTLR2 and ENA-78 (r = 0.39, *p* = 0.001), sTLR2 and GRO-α (r = 0.63, *p* < 0.0001), sTLR2 and I-TAC (r = 0.79, *p* < 0.0001), sTLR2 and Fractalkine (r = 0.48, *p* < 0.0001), sTLR2 and SDF-1α (r = 0.24, *p* = 0.043), and sTLR2 and MIP-3α (r = 0.77, *p* < 0.0001). There was no significant correlation between sTLR2 and other chemokines tested in SIV-infected RhMs.

### 3.7. Absence of Correlation between Plasma sTLR2 and Mucosal Inflammatory Markers

The correlation between plasma sTLR2 and different mucosal inflammatory markers was analyzed to determine whether the increased sTLR2 level is associated with intestinal inflammation. We were unable to detect any correlation between sTLR2 and REG3A (r = 0.12, *p* = 0.397), sTLR2 and IFABP (r= −0.15, *p* = 0.232), and sTLR2 and sCD14 (r = −0.03, *p* = 0.803) (Figure 4F). The mean ± SE values of REG3A, IFABP, and sCD14 ranged from 74.9 ± 8.2 to 131.6 ± 57.5 pg/mL, 62.1 ± 29.2 to 393.1 ± 149.8 pg/mL, and 4,092,821 ± 312,403 to 8,802,893 ± 1,205,605 pg/mL, respectively. Our longitudinal study of SIV infection did not reveal any significant changes in the plasma sCD14 levels [33].

### 3.8. PBMC or Jejunum LPL NK, B, and CD8+ T Cells mb-TLR2 Expression Was Significantly Upregulated after Infection

Staphylococcal enterotoxin B has been found to induce upregulation of TLR2 in monocytes [51]. When stimulated by SEB, all cell populations, including CD8+ T cells, B cells, NK cells, CD4+ cells, and monocytes, showed a significant increase in TLR2 expression compared to the media controls (Figure 5). This suggests that SEB can cause the upregulation of TLR2 in various cell populations. The frequency of mb-TLR2 expression on CD8+ (CD3+CD14–CD8+) T cells, CD20+ B cells, NK (CD3–CD14–CD20–CD8+), monocytes (CD3–CD14+), and CD4+ (CD3+CD8–) T cells from PBMC and jejunum LPL pre- and post-SIV infection was analyzed using multicolor flow cytometry assay (Figure 5). A significant increase in mb-TLR2 expression was detected in CD8+, CD20+, and NK cells isolated from PBMC during SIV infection when compared to their preinfection time point. The mb-TLR2-positive CD8+ T cells were significantly higher at all the post-infection time points (40 dpi: 9.3 ± 1.5%, *p* = 0.014; 60 dpi: 5.4 ± 0.8%, *p* = 0.015; 70 dpi: 3.9 ± 0.5%, *p* = 0.016; 90 dpi: 4.0 ± 0.3%, *p* = 0.001; 120 dpi: 6.8 ± 1.2%, *p* = 0.041; 150 dpi: 8.4 ± 1.4%, *p* = 0.015; 180 dpi: 5.0 ± 0.7%, *p* = 0.014) except at 14 and 21 dpi compared to day 0 (1.5 ± 0.2%) (Figure 6A,B). Similarly, the mb-TLR2 expression in CD20+ B cells was also significantly higher at 21 dpi (8.8 ± 1.3%, *p* = 0.016), 150 dpi (15.4 ± 2.4%, *p* = 0.006), and 180 dpi (12.7 ± 2.3%, *p* = 0.023) compared to the day 0-time point (1.9 ± 0.2%) (Figure 6A,C). Higher mb-TLR2 expression was noticed in PBMCs CD8+ and CD20+ cells throughout the infection. An increased mb-TLR2 expression trend was also detected in NK cells isolated from PBMCs after infection. A significant increase in mb-TLR2+ NK cells was observed at 40 (3.9 ± 0.6%, *p* = 0.031) and 60 dpi (3.0 ± 0.6%, *p* = 0.048) compared to the preinfection time point (1.0 ± 0.1%) (Figure 6A,D). We could not observe significant differences in mb-TLR2 expression from PBMC CD4 and monocyte populations during the infection (Appendix A). A correlation analysis investigated the relationship between plasma sTLR2 and mb-TLR2 in different PBMC cell subsets. The study revealed a positive correlation between plasma sTLR2 and mb-TLR2+CD20+ B cells (r = 0.336, *p* = 0.005), as well as mb-TLR2+NK cells from PBMCs (r = 0.296, *p* = 0.015). There was no correlation between plasma sTLR2 level and any other mb-TLR2+cell population in PBMCs.

All SIV-infected RhMs showed significant loss of CD4+ T cells in the jejunum by 21 dpi. SIV infection resulted in a significant upregulation of mb-TLR2+ cells in jejunum NKs. We noted a significant increase in the frequency of mb-TLR2+NK cells at all three post-infection time points (21 dpi: 12.6 ± 1.2%, *p* = 0.008; 90 dpi: 11.5 ± 0.9%, *p* = 0.004; 180 dpi: 10.9 ± 1.3%, *p* = 0.015) compared to the 0 day time point (5.5 ± 0.8%) (Figure 6E). Even though the frequency of mb-TLR+ B cells is higher in chronic infections (90 dpi: 9.4 ± 2.6%; 180 dpi: 10.8 ± 2.0%), compared to the day 0-time point (6.9 ± 1.8%), no statistically significant difference was observed among all those time points (Appendix A). No changes in mb-TLR2+ cells were detected in the Jejunum CD8+ T cell population (Appendix A). We could not perform mb-TLR2 analysis on jejunum monocytes as they were absent in our isolated cells. Due to substantial CD4+ T cell loss during infection, mb-TLR2 expression in jejunum CD4+ T cells was excluded from our analysis. No correlation was found between plasma sTLR2 level and any other mb-TLR2+ cell population in the jejunum LPL.

We observed a higher frequency of mb-TLR2-expressing B cells, CD8+ T cells, and NK cells in the LN at all post-infection time points. However, no statistically significant changes were noted, as shown in Appendix A. Analysis of mb-TLR2+ monocytes from LN was excluded as the frequency of monocytes was very low.

Since FLT3L plays a role in the growth and development of NKs, we quantified NK populations in both PBMCs and jejunum LPLs. Quantification of absolute NK cells in circulation showed no significant changes in the number of NKs at different time points after infection, but a slightly higher count was observed at 21 dpi (821 ± 216/µL of blood) when compared to the pre-infection time point (429 ± 104/µL of blood) (Figure 7A). A similar pattern in the frequency of NK cells in the jejunum LPLs was observed. The higher NK cell frequency at 21 dpi (20.4 ± 2.1%, *p* = 0.029) was significantly different from the pre-infection level (12.3 ± 1.5%) (Figure 7B). However, the NK cell frequency returned to normal at 90 and 180 dpi compared to the baseline (Figure 7B). We also investigated whether there was a correlation between the NK cells and plasma FLT3L concentration. Conversely, no correlation was detected between the two in the circulation (r = −0.096, *p* = 0.45) and jejunum LPL (r = −0.142, *p* = 0.40).

### 3.9. TLR2 Agonist Modulates Cytokine/Chemokine Production In Vitro

We conducted experiments using PBMCs from three SIV-uninfected, normal RhMs to investigate mb-TLR2’s role in regulating cytokine and chemokine production in cells. In vitro cultures in the presence of protein A, a TLR2 agonist, showed a significantly increased production of IL-1RA (mean, 3166.7 versus 944.3 pg/mL, *p* = 0.048), IL-9 (mean, 0.2 versus 0.0 pg/mL, *p* < 0.0001), IL-15 (mean, 1.5 versus 0.5 pg/mL, *p* = 0.041), Gro-α (mean, 2719.1 versus 47.6 pg/mL, *p* = 0.008), and ENA-78 (mean, 165.0 versus 5.6 pg/mL, *p* = 0.017) compared to media controls (Figure 8A–E). These results suggest that increased mb-TLR2 expression might have caused increased production of IL-1RA, IL-9, IL-15, GRO-α, and ENA-78 in SIV-infected RhMs (Figure 4D,E).

Protein A-mediated TLR2 stimulation also significantly upregulates IL-1A (mean, 0.4 versus 0.0 pg/mL, *p* = 0.003), IL-8 (mean, 9.6 versus 0.04 pg/mL, *p* = 0.023), IL-17A/F (mean, 719.8 versus 2.5 pg/mL, *p* = 0.008), IL-17B (mean, 0.3 versus 0.0 pg/mL, *p* = 0.006), IL-17C (mean, 2.6 versus 0.0 pg/mL, *p* = 0.031), I-309 (mean, 1.1 versus 0.0 pg/mL, *p* = 0.010), MCP-4 (mean, 9.2 versus 1.2 pg/mL, *p* = 0.040), MIP-1β (mean, 68.2 versus 2.5 µg/mL, *p* = 0.0002), MIP-5 (mean, 0.94 versus 0.91 pg/mL, *p* = 0.003), and YKL-40 (mean, 5.2 versus 4.7 pg/mL, *p* = 0.031) compared to medial control (Figure 8F–O). This implies that activation of mb-TLR2 can upregulate various cytokines and chemokines, including IL-1A, IL-8, IL-17A/F, IL-17B, IL-17C, I-309, MCP-4, MIP-1β, MIP-5, and YKL-40. However, we were unable to detect any significant correlation between sTLR2 and plasma IL-1A, IL-8, IL-17A/F, IL-17B, IL-17C, I-309, MCP-4, MIP-1β, MIP-5, and YKL-40 cytokines/chemokines in SIV-infected subjects. During the assay, several other cytokines and chemokines were evaluated. However, no significant changes were detected in the protein A-stimulated culture compared to the media control cells.

## 4. Discussion

Dysregulation of the adaptive immune system is well documented as the cause of HIV disease progression. Nevertheless, the innate immune system is the first line of defense against microbial infections and plays a crucial role in responding to them. Host TLRs recognize viral proteins as they trigger rapid antiviral responses and shape adaptive immunity. TLR2 detects bacterial lipoproteins by forming complexes with TLR1 and TLR6 [52]. Studies have found that TLR2 gene expression increased significantly in untreated chronic HIV-1 and AIDS patients’ PBMCs compared to treated and uninfected individuals [53]. During early or late HIV-1 exposure, TLR2 mRNA expression was differentially expressed in in vitro PBMC culture. The triggering of TLR2 primarily accelerates the production of new HIV-1 in immature monocyte-derived dendritic cells by the induction of the NF-κB signaling pathway [30,54]. Our study establishes the dynamics of sTLR2 in plasma and mb-TLR2 expression in peripheral and mucosal tissues, providing strong evidence of a positive correlation between sTLR2 and various cytokines and chemokines for the first time. Additionally, a proof-of-concept study demonstrated that activation of mb-TLR2 can directly increase the expression of specific pro-inflammatory cytokines observed during HIV infection using a macaque model. Our findings suggest that TLR2 plays an essential role in the development of HIV-1 disease.

After HIV infection, activation of TLRs can lead to varying responses depending on the involved cell type and cytokine/chemokine. TLR2 can also impact cytokine/chemokine production directly or indirectly. The positive correlation between sTLR2 and various cytokines (IL-18, IL-1RA, IL-15, IL-13, IL-9, TPO, FLT3L, and IL-17F) and chemokines (IP-10, MCP-1, MCP-2, ENA-78, GRO-α, I-TAC, Fractalkine, SDF-1α, and MIP-3α) suggests that TLR2 plays a role in modulating HIV immunity and pathogenesis. The positive correlation between IL-18 and PVL corroborates a prior report where IL-18 has been found to promote HIV replication by inducing CXCR4 co-receptor expression in the PBMCs from HIV-1-infected patients receiving antiretroviral therapy [41]. IL-18 also serves various functions, including IFNγ induction, NK cell activity reinforcement, cytotoxic function enhancement, and promoting inflammatory responses and autoimmune diseases [55]. The precise mechanism behind the significantly increased expression of IL-18 during acute and chronic phases of SIV infection remains incompletely understood. The absence of IL-18 production in TLR2 knockout primary bone marrow-derived macrophages treated with SARS-CoV-2 E protein and ATP [56] indicates that TLR2 is directly involved in regulating IL-18 expression during SIV/HIV infection. The increased MCP-1 may also drive the recruitment of CXCR4 expression on resting CD4+ T cells, favoring increased viral replication in HIV-1-infected patients [57]. Elevated levels of sTLR2 had also been reported in cerebrospinal fluid (CSF) of SIV-infected macaques with neurological sequelae compared to those without neurological complications. A positive correlation was observed between CSF sTLR2 and MCP-1 concentration, further emphasizing the role of sTLR2 in developing SIV-associated neuroinflammation and subsequent neuropathology [58]. We observed increasing mb-TLR2 expression in NK cells and decreasing CD4+ T cells in the jejunum following infection. There was also an increase in plasma IL-15 production, which positively correlated with the plasma sTLR2. Under normal conditions, IL-15 production was believed to be responsible for differentiating intraepithelial lymphocytes and regulating intestinal homeostasis through TLR2-dependent mechanisms [59]. However, in cases of inflammatory bowel disease and celiac disease, IL-15 production is increased, leading to the recruitment and activation of inflammatory and innate immune cells and the production of IFN-γ and TNF-α. Our previous study has shown that IFN-γ and TNF-α are upregulated in intestinal lamina propria cells during SIV infection [34]. IL-15 is also upregulated, likely due to the upregulation of TLR2 in these subjects. This overexpression of IL-15 is crucial for the induction of mucosal inflammation, and further research is necessary to comprehend TLR2-mediated IL-15 regulation in the context of HIV infection. On the contrary, breast milk obtained from HIV-infected women exhibits a significantly higher concentration of sTLR2 compared to HIV-uninfected milk. The elevated sTLR2 level is positively correlated with IL-15, indicating that a local innate compensatory mechanism may play a role in combating HIV infection in infants [60].

Monocytes play a critical role in the early immune response to infection by directly recognizing PAMPs through their higher expression of mb-TLR2, which is consistent with previous findings [28,61]. Conversely, the absence of significant changes in mb-TLR2 expression in monocytes and the absence of sCD14 upregulation in our study suggest that monocyte activation might have a limited impact on the outcome of this study. Furthermore, the lack of correlation between sCD14 and sTLR2 supports the idea that the upregulation of sTLR2 might not be linked to monocyte activation. Instead, it might be associated with activating peripheral or mucosal cytotoxic CD8+ T cells, B cells, and NK cells during HIV infection. In contrast, a higher plasma sCD14 was detected in HIV-infected and ART-treated patients when compared to age-matched HIV-uninfected controls [62]. REG3A, a protein expressed only in the intestine and pancreas, plays a crucial role in regulating the interaction between the host and the microbiota and controlling the inflammation of the mucous membrane [63].

Similarly, I-FABP, a marker of enterocyte damage or death [62], was also found to have no significant correlation with sTLR2 levels. No significant difference in I-FABP levels was also reported between HIV-infected, ART-treated patients, and age-matched HIV-uninfected controls earlier [62]. sTLR2 is a protein that is produced by shedding the ectodomain of mb-TLR2. It has been found in various body fluids such as breast milk, plasma [64], saliva [65], and amniotic fluid [66,67]. Increased sTLR2 binding with PAMPs and CD14 reduces mb-TLR2 activation, preventing its binding with CD14 and PAMPs. This suggests that sTLR2 acts as a negative regulator in preventing mb-TLR2 activation, as reported previously [11,68]. HIV-1 structural proteins like p17, p24, and gp41 act as PAMPs and cause immune activation by the NFκB signaling pathway through TLR2 [54]. Increased TLR2 mRNA expression was detected in monocytes isolated from HIV-1-infected patients [24]. TLR2 mRNA expression, as well as TLR3 and TLR4, were found to be elevated in patients with advanced disease and chronic viremia [53]. Our study has revealed that during HIV infection, there is an activation of TLR2, as evidenced by the increased upregulation of sTLR2 and mb-TLR2. As previously reported, the plasma sTLR2 upregulation may have a compensatory mechanism to reduce mb-TLR2 activation. However, the positive correlation between sTLR2 and PBMC mb-TLR2+CD20+ B cells and mb-TLR2+ NK cells suggests that the increase in sTLR2 levels may have resulted from the fragmentation of NK and B cell mb-TLR2, which needs additional future study. Our research suggests that identifying the loss of epithelial ZO-1 and E-cadherin expression and the upregulation of mucosal IL-10 and TGF-β+ cells are crucial markers for detecting mucosal inflammation [3,8,34,69], as opposed to relying on plasma measurements of sCD14, I-FABP, and REG3A levels, which we have presented here.

The significant positive correlation between sTLR2 and anti-inflammatory cytokines, such as IL-1RA and IL-13, suggests that sTLR2 plays a protective and immunoregulatory role in regulating HIV infection. Increased production of IL-1RA inhibits IL-1α- and IL-1β-mediated cellular activation [70], while increased IL-13 inhibits the function of monocytes and macrophages, which are responsible for the production of TNF, IL-1, IL-8, and MIP-1α. Increased IL-13 production has been associated with reduced viral load and decreased cytokine-favored disease progression [71]. The positive correlation between sTLR2 and FLT3L also highlights the protective function of sTLR2 in augmenting the dendritic cell population in combating HIV infection. A study conducted on HIV-humanized NSG and BLT mice showed that FLT3L administration triggered the expansion and mobilization of plasmacytoid dendritic cells during HIV infection, which reduced viremia [72]. Nonetheless, the study also found that the plasmacytoid dendritic cells generated by FLT3L were more responsive to TLR7 stimulation [72].

TLR2 plays a dual role in infection processes. On the one hand, TLR2 is essential for releasing cytokines and chemokines and inducing immune activation. On the other hand, TLR2 has been shown to have a protective role by promoting the production of IL-1R, IL-13, or IL-15. This study shows a positive correlation between most cytokines and chemokines with sTLR2 levels, suggesting that TLR2 significantly regulates immune activation and disease-mediated inflammation. Certain viral structural proteins, such as measles hemagglutinin A, cytomegalovirus glycoprotein B, hepatitis C virus core, and herpes simplex virus-1 glycoprotein H/L and B proteins, can induce the production of inflammatory cytokines via the TLR2 sensing pathway. TLR2 can also trigger type I interferon responses by a specific subset of inflammatory monocytes in response to the viral ligands. In mice, the administration of Pam_2_Cys, a TLR2 agonist, has been shown to protect against lethal influenza A infection [73]. Similarly, in a SARS-CoV-2 ferret model, prophylactic intra-nasal administration of INNA-051, a TLR2/6 agonist, significantly reduced nose and throat SARS-CoV-2 RNA levels [74]. On the contrary, a recent study shows that blocking TLR2 signaling with ox-PAPC can protect from severe SARS-CoV-2 pathogenesis in the K18-ACE2 transgenic mice model [56]. SMU-Z1, a potential TLR1/2 agonist, can enhance antiviral activity by augmenting NK cell-mediated antiviral responses and reversing HIV-1 latency indirectly, both in in vitro and ex vivo experiments [75]. Therefore, a potential solution to regulate immune activation during HIV infection could be to develop a therapy that explicitly governs the production of both sTLR2 and mb-TLR2 expression. It is essential to carefully consider the timing and regulation of TLR2 signaling to avoid delaying viral clearance or accelerating tissue damage and immune activation. Future studies are necessary to understand the potential of TLR2 blockers or agonists in regulating immune activation and cytokine/chemokine regulation in the RhM-SIV model. Despite some limitations in our study, such as the inability to measure the expression of TLR2 in different cells, including macrophages and epithelial cells, at various time points and the lack of measurement of the LPS binding protein, our study had a significant strength. We collected and tested longitudinal samples from the jejunum, plasma, lymph nodes, and peripheral blood mononuclear cells throughout the SIV infection.

## 5. Conclusions

Our research has revealed that TLR2 is critical in regulating the body’s inflammatory response during HIV infection. The presence of mb-TLR2 on peripheral NK and B cells is positively associated with the induction of sTLR2 production. Therefore, sTLR2 can be used as a potential biomarker to determine disease progression and general immune activation in HIV infection. Through regulating proinflammatory and anti-inflammatory cytokines and chemokine responses, the TLR2 pathway controls systemic inflammatory reactions. Our study provides novel insights into the immune system’s response to HIV, which may ultimately result in the development of new therapeutic strategies targeting TLR2 expression during HIV infection.

## Figures and Tables

**Figure 1 vaccines-11-01861-f001:**
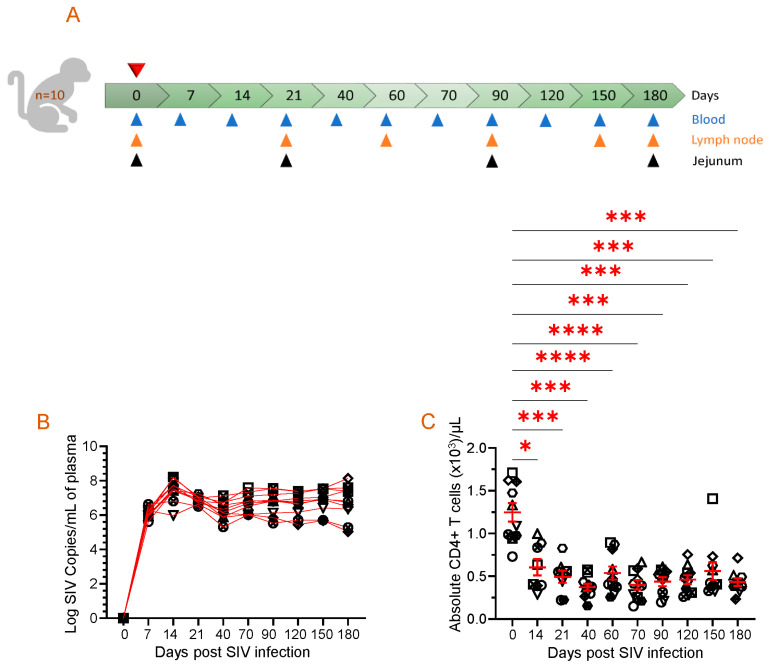
Animal study design, viral RNA quantification, and absolute CD4 count in the periphery are displayed. (**A**) A schematic of the study where 10 macaques were infected with pathogenic SIV and samples were collected at the indicated time points is shown. Each triangle with a different color represents time points and tissue collected for performing assays. The top red triangle denotes that the animals were infected with pathogenic SIV at day 0. (**B**) Plasma viral load (PVL) for each SIV-infected macaque is demonstrated for the duration of the study, as determined by RT-PCR (*n* = 10). Each symbol represents the PVL per mL of plasma for an individual macaque. (**C**) The data presented in this study highlight the absolute peripheral CD4+ T cell count before and after SIV infection. Each symbol on the scattered plot demonstrates the information gathered from individual macaques. The red line with error bars represents the mean ± SE for each time point. Statistical significance was tested using Tukey’s multiple comparison test (*, *p* ≤ 0.05; ***, *p* ≤ 0.001; ****, *p* ≤ 0.0001).

**Figure 2 vaccines-11-01861-f002:**
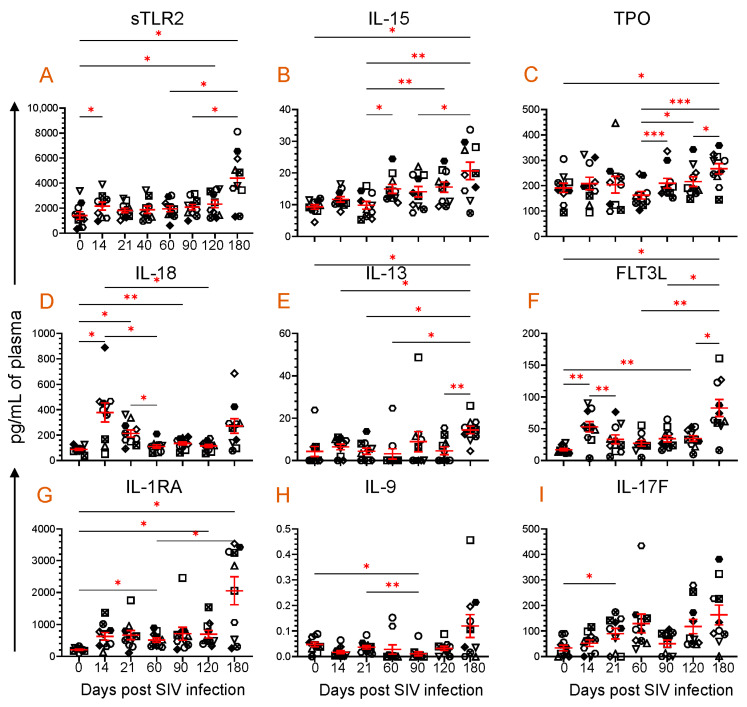
Plasma sTLR2 and eight cytokine profiles during SIV infection. Scattered plots with mean ± SE for (**A**) sTLR2 and eight different cytokines ((**B**), IL-15; (**C**), TPO; (**D**), IL-18; (**E**), IL-13; (**F**), FLT3L; (**G**), IL-1RA; (**H**), IL-9; and (**I**), IL-17F) responses observed throughout the SIV infection time points are shown. Each symbol on the scattered plot represents the cytokine concentration for an individual macaque. The red line with bars represents mean ± SE for each time point. Asterisks indicate statistical differences between time points as calculated by Tukey’s multiple comparison analysis (*, *p* ≤ 0.05; **, *p* ≤ 0.01; ***, *p* ≤ 0.001).

**Figure 3 vaccines-11-01861-f003:**
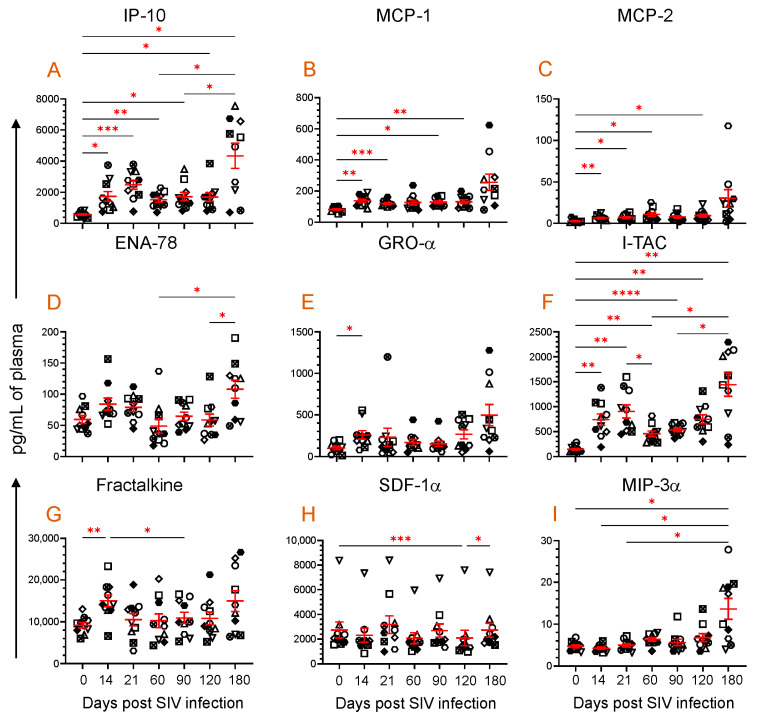
Plasma chemokine profile during SIV infection. Scattered plots with mean ± SE for nine different chemokines ((**A**), IP-10; (**B**), MCP-1; (**C**), MCP-2; (**D**), ENA-78; (**E**), GRO-α; (**F**), I-TAC; (**G**), Fractalkine; (**H**), SDF-1α; and (**I**), MIP-3α) responses observed throughout the SIV infection time points are shown. Each symbol of the scattered plot represents the chemokine concentration for an individual macaque. The red line with bars represents mean ± SE for each time point. Asterisks indicate statistical differences between time points as calculated by Tukey’s multiple comparison analysis (*, *p* ≤ 0.05; **, *p* ≤ 0.01; ***, *p* ≤ 0.001; ****, *p* ≤ 0.0001).

**Figure 4 vaccines-11-01861-f004:**
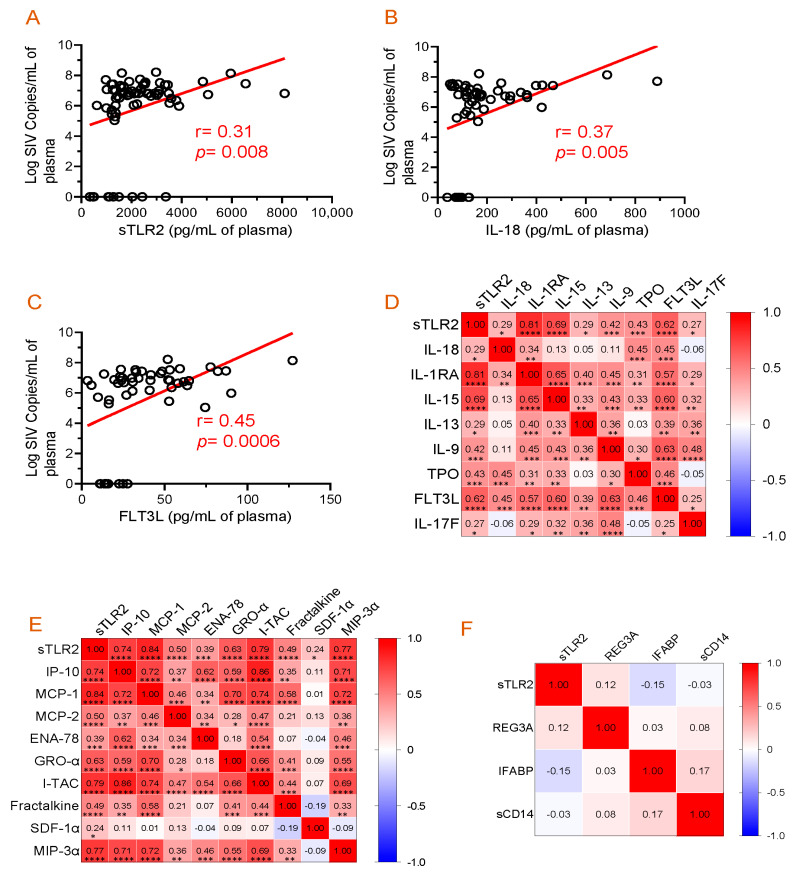
Correlation analysis among plasma viral load, cytokines, chemokines, and inflammatory markers. (**A**–**C**) Significant positive correlations between plasma viral loads and sTLR, IL-18, or FLT3L were detected. Correlation matrix of sTLR2 and cytokines (**D**) and sTRL2 and chemokines (**E**) at D0, 14, 21, 60, 90, 120, and 180 post-SIV infection time points in SIV-infected macaques. (**F**) Correlation matrix of sTLR2 and inflammatory markers (REG3A, I-FABP, and sCD14) at D0, 14, 21, 60, 90, 120, and 180 post-SIV infection time points in SIV infected macaques. Two-tailed Pearson correlation values (r) are shown from red (1.0, positive correlation) to blue (–1.0, negative correlation). In each box, r values indicate positive and negative correlations by number and color. Asterisks show significant differences (*, *p* ≤ 0.05; **, *p* ≤ 0.01; ***, *p* ≤ 0.001; ****, *p* ≤ 0.0001).

**Figure 5 vaccines-11-01861-f005:**
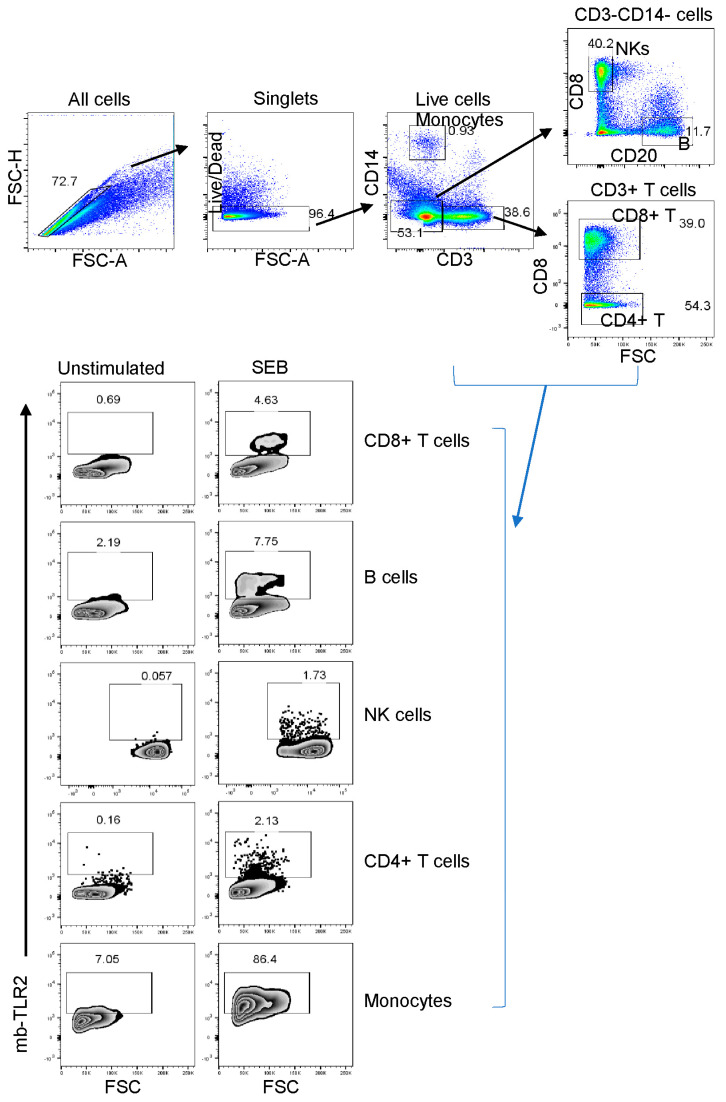
The TLR2 expression of various cell subsets in a representative macaque’s peripheral blood mononuclear cells was detected using flow cytometry before infection. First, the cells were gated to identify singlets and then live cells. After that, CD3+ or CD14+ monocytes were identified. The CD3–CD14– cells were then gated further to identify B cells and NK (CD3–CD14–CD20–CD8+) cells based on their CD8 and CD20 expression. CD3+CD8– T cells were defined as CD4+ T cells in this study. The study evaluated the TLR2 expression of various cell subpopulations, showing the percentage of each boxed-gated cell population for unstimulated media control and *Staphylococcus enterotoxins B* (SEB) stimulated cells. It’s important to note that SEB causes cell activation and increases the presence of mb-TLR2-positive cells in the culture.

**Figure 6 vaccines-11-01861-f006:**
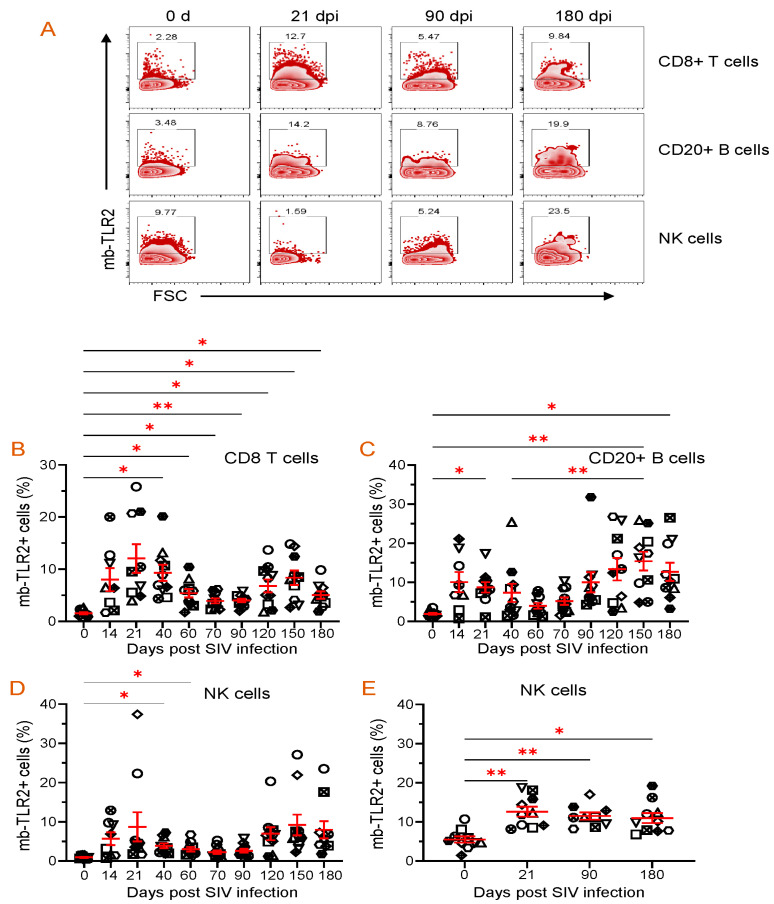
Dynamics of mb-TLR2 expression of different cell subsets during SIV infection in PBMC and jejunum lamina propria lymphocytes. (**A**) Representative expression of mb-TLR2 in PBMC from a macaque at different time points after SIV infection. Percentages of mb-TLR2+ cells in either CD8+ T cells, CD20+ B cells, or NK cells are shown in the box of each zebra plot. The frequency of mb-TLR2+ cells in (**B**) CD8+ T cells, (**C**) CD20+ B cells, and (**D**) NK cells in PBMC is shown for SIV-infected macaques over the infection time points. (**E**) Jejunum lamina propria mb-TLR2+ NK cells also significantly increased following SIV infection. Red lines with errors show mean ± SE for each time point. Each symbol on the scattered plot represents the percentage of mb-TLR2+ cells for an individual macaque. Asterisks indicate statistical differences between time points calculated by Tukey’s multiple comparison analysis (*, *p* ≤ 0.05; **, *p* ≤ 0.01).

**Figure 7 vaccines-11-01861-f007:**
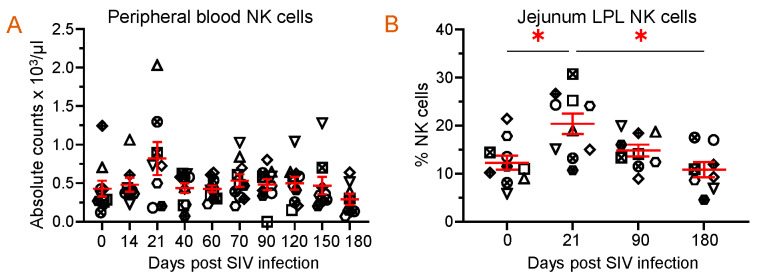
Dynamics of NK population during SIV infection in peripheral blood and jejunum lamina propria lymphocytes. (**A**) Absolute count of NK cells in peripheral blood and (**B**) frequency of NK cells in jejunum lamina propria lymphocytes are shown for SIV-infected animals over time (*n* = 10). Red lines with errors show mean ± SE for each time point. Each symbol on the scattered plot represents the frequency of NK cells for an individual macaque. Asterisks indicate statistical differences between time points calculated by Tukey’s multiple comparison analysis (*, *p* ≤ 0.05).

**Figure 8 vaccines-11-01861-f008:**
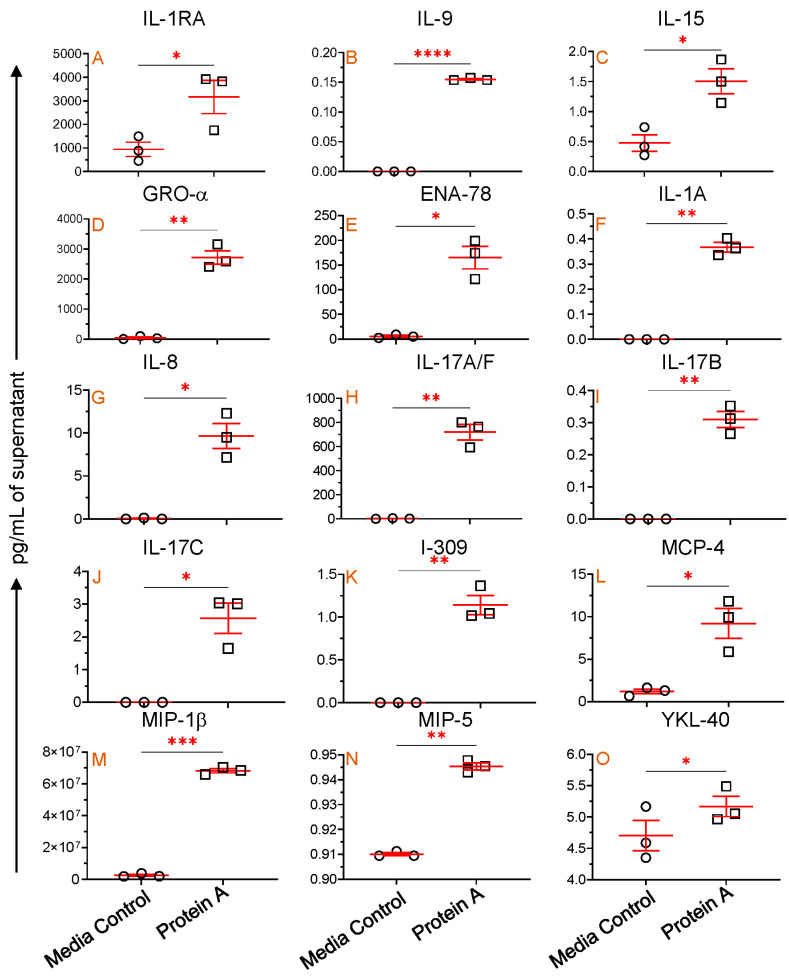
mb-TLR2-mediated cytokine and chemokine regulation. IL-1RA (**A**), IL-9 (**B**), IL-15 (**C**), GRO-α (**D**), and ENA-78 (**E**) concentrations were detected in the cell culture supernatant with and without Protein A stimulation. An additional 10 different cytokines and chemokines ((**F**), IL-1A; (**G**), IL-8; (**H**), IL-17A/F; (**I**), IL-17B; (**J**), IL-17C; (**K**), I-309; (**L**), MCP-4; (**M**), MIP-1β; (**N**), MIP-5, and (**O**), YKL-40) were also significantly increased in the cell culture supernatant after Protein A stimulation compared to unstimulated controls. Red lines with errors show mean ± SE for each time point. The asterisks indicate significant differences between media control and Protein A-stimulated wells (*n* = 3) as calculated by paired *t*-test analysis (*, *p* ≤ 0.05; **, *p* ≤ 0.01; ***, *p* ≤ 0.001; ****, *p* ≤ 0.0001). The circle and rectangle symbols represent media control and protein A-stimulated cultures respectively.

## Data Availability

All relevant data are included within the manuscript. The raw data are available on request from the corresponding author.

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
