# Peer review of "Toll-like Receptor 2 Mediated Immune Regulation in Simian Immunodeficiency Virus-Infected Rhesus Macaques"

_vaccines, 2023, doi:10.3390/vaccines11121861_

Round 1
Reviewer 1 Report
Comments and Suggestions for Authors
This manuscript investigated the level of HIV-1 viral load in plasma, sTLR 2, and various cytokines with infection time in the SIV-infected rhesus macaque model and found that TLR 2 is crucial to regulate the body's inflammatory response during HIV-1 infection. TLR 2 regulates the systemic inflammatory response by regulating the pro-inflammatory and anti-inflammatory cytokines and chemokines. The article is clearly expressed, logical, and reasonable and clarifies critical scientific issues. Some minor issues need to be explained:
1. Why did the sTLR2 levels increase significantly on Day 180? What are the possible reasons?
2. Plasma IL-18 levels were highest at 14 days of infection, subsequently decreased, and then increased at 180 days. The r between sTLR2 and IN-18 was 0.29, which does not imply a strong correlation. What is the possible reason for IL-18 enhanced on day 14 and day 180?
3. FLT3L plays an essential role in the growth and development of NK cells, which showed a similar trend with IL-18. What is the number of NK cells during SIV infection?
4. In Figure 6, panels D and E looked the same, and I suggest the authors modify the ordinate.
5. What is TLR 2 expression in PBMC stimulated with protein A and without protein A in a PBMC cell model of SIV infection?
6. According to the dynamic changes of different cytokines in this study, how can we better respond to HIV-1 infection by regulating TLR2? The author can discuss it appropriately.
Author Response
This manuscript investigated the level of HIV-1 viral load in plasma, sTLR 2, and various cytokines with infection time in the SIV-infected rhesus macaque model and found that TLR 2 is crucial to regulate the body's inflammatory response during HIV-1 infection. TLR 2 regulates the systemic inflammatory response by regulating the pro-inflammatory and anti-inflammatory cytokines and chemokines. The article is clearly expressed, logical, and reasonable and clarifies critical scientific issues.
Response: We appreciate your positive feedback. The revised text now includes all the corrections requested by the reviewer. We hope that our responses have taken care of your concerns.
Some minor issues need to be explained:
- Why did the sTLR2 levels increase significantly on Day 180? What are the possible reasons?
Response: We observed a significant upregulation of sTLR2 as early as 14 days post-infection (dpi), and the levels remained high throughout the study period. We also noted a significant increase at 120 and 180 dpi (refer to page 7, lines 279-285). In our discussion section (refer to page 20, lines 615-625), we have included a possible explanation for this upregulation.
- Plasma IL-18 levels were highest at 14 days of infection, subsequently decreased, and then increased at 180 days. The r between sTLR2 and IN-18 was 0.29, which does not imply a strong correlation. What is the possible reason for IL-18 enhanced on day 14 and day 180?
Response: Our manuscript acknowledges the reviewer's comment and confirms that we do not assert a highly significant correlation between sTLR2 and IL-18. However, we have now included a discussion of our positive correlation between sTLR2 and IL-18, referencing recent publications, in the discussion section (page 19, lines 561-570).
- FLT3L plays an essential role in the growth and development of NK cells, which showed a similar trend with IL-18. What is the number of NK cells during SIV infection?
Response: We would like to acknowledge the reviewer for their valuable comment. We have reanalyzed the NK population in peripheral blood and gut tissue and included a new Figure 7 in our report. Our findings suggest no significant upregulation in the NK population in the peripheral blood. However, the gut NK population increased significantly during acute infection compared to the baseline. We also conducted a correlation analysis between serum FLT3L and tissue NK population but found no correlation. Our results are detailed in the result section on page 14, lines 465-476.
- In Figure 6, panels D and E looked the same, and I suggest the authors modify the ordinate.
Response: We have revised those panels in the revised Figure 6.
- What is TLR 2 expression in PBMC stimulated with protein A and without protein A in a PBMC cell model of SIV infection?
Response: We did not measure sTLR2 and mbTLR2 expression in protein A-stimulated PBMCs from SIV-infected animals in this study.
- According to the dynamic changes of different cytokines in this study, how can we better respond to HIV-1 infection by regulating TLR2? The author can discuss it appropriately.
Response: We thank the reviewer for providing this valuable feedback. We want to confirm that we have addressed the role of TLR2 regulation in controlling HIV-1 pathogenesis in our discussion section (on pages 20-21, lines 643-666).
Reviewer 2 Report
Comments and Suggestions for Authors
Nongthombam Boby et al showed a positive correlation of soluble TLR2 and membrane bound TLR2 with many cytokines and chemokines, blood and in cells isolated from PBMCs and jejunum after infection with SIV in macaques. Authors are trying to establish the positive correlation of sTLR2 and mbTLR2 with various cytokines and chemokines for the first time. Though this study shows a positive correlation of sTLR with cytokine and inflammatory markers, there is no experimental proof for sTLR2 being causative for the upregulation of cytokines and/or inflammatory cytokines. Either injection of recombinant soluble TLR2 or full length TLR2 into macaques or treating the PBMCs from control macaques are needed to establish the role of upregulating cytokines and inflammatory signatures by TLR2. While protein A, a ligand for TLR induced cytokine enhancement is supportive, the causative role of sTLR2 require additional work.
- Authors should also comment on previous studies showing the presence of sTLR2 in brain in HIV infected patients and macaques (PMID: 27882497); and correlation of sTLR2 and IL15 (PMID: 25265071).
- Authors may explain to the editors why they are abstaining from showing the data on lower CD4+T cells as they are markers of disease progression.
- In figure 7, the statistical significance on IL9 (A), IL1A, IL17B, MIP1B, MIP-5 and YKL-40 (B) do not appear visually significant on the graph and may consider reevaluating or plotting differently.
- Under method section, more details on the qPCR method is required: primer sequence, standard (if any) information and details of how the viral copies are calculated.
some minor edits are needed
Author Response
Nongthombam Boby et al showed a positive correlation of soluble TLR2 and membrane bound TLR2 with many cytokines and chemokines, blood and in cells isolated from PBMCs and jejunum after infection with SIV in macaques. Authors are trying to establish the positive correlation of sTLR2 and mbTLR2 with various cytokines and chemokines for the first time. Though this study shows a positive correlation of sTLR with cytokine and inflammatory markers, there is no experimental proof for sTLR2 being causative for the upregulation of cytokines and/or inflammatory cytokines. Either injection of recombinant soluble TLR2 or full length TLR2 into macaques or treating the PBMCs from control macaques are needed to establish the role of upregulating cytokines and inflammatory signatures by TLR2. While protein A, a ligand for TLR induced cytokine enhancement is supportive, the causative role of sTLR2 require additional work.
Response: We are grateful for your positive feedback. Several studies have shown that the upregulation of TLR2 expression is associated with increased immune activation through the upregulation of various cytokines and chemokines. In our report, we have highlighted the detection of upregulation of both sTLR2 and mbTLR2 during SIV infection. Furthermore, the upregulation of sTLR2 may be a compensatory mechanism that reduces mb-TLR2 activation. Additionally, we have found a positive correlation between sTLR2 and PBMC mb-TLR2+CD2+ B cells and mb-TLR2+Nk cells. We agree with the reviewer's suggestion that an in vivo experiment would be ideal for exploring the role of sTLR2 in regulating inflammatory cytokines in HIV infection models. The revised text now incorporates all the requested corrections from the reviewer. We hope that our responses have satisfactorily addressed your concerns.
Additional comments:
- Authors should also comment on previous studies showing the presence of sTLR2 in brain in HIV infected patients and macaques (PMID: 27882497); and correlation of sTLR2 and IL15 (PMID: 25265071).
Response: We have included the studies you suggested in our discussion on page 19, lines 573-577 and 589-593.
- Authors may explain to the editors why they are abstaining from showing the data on lower CD4+T cells as they are markers of disease progression.
Response: We apologize for not including the CD4 data in our initial submission. We have now included the CD4 data as Figure 1C in this revised manuscript.
- In figure 7, the statistical significance on IL9 (A), IL1A, IL17B, MIP1B, MIP-5 and YKL-40 (B) do not appear visually significant on the graph and may consider reevaluating or plotting differently.
Response: Thank you for your input. We've revised Figure 8 accordingly.
- Under method section, more details on the qPCR method is required: primer sequence, standard (if any) information and details of how the viral copies are calculated.
Response: We have revised the method section accordingly (page 4, lines 122-136).
Round 2
Reviewer 2 Report
Comments and Suggestions for Authors
Authors have addressed all my suggestions and I don't have any further comments.
Comments on the Quality of English LanguageAuthors have addressed all my suggestions and I don't have any further comments.